# Exploring the Linguistic and Cultural Identities of Transnational Background Children in Catalonia, Spain [†]

Claudia Vallejo Rubinstein * and Valeria Tonioli *

Department of Translation, Interpreting and East Asian Studies, Universitat Autònoma de Barcelona, 08193 Bellaterra, Spain
* Correspondence: claudia.vallejo@uab.cat (C.V.R.); valeria.tonioli@uab.cat (V.T.)
† NEW ABC project.

**Abstract:** This article explores linguistic and cultural identities as they emerge in ethnographic data from plurilingual children with transnational and ethnic minority backgrounds in Catalonia, Spain. The particular sociolinguistic and multicultural context where these young people currently live, characterised by the coexistence of local, national and heritage languages with unequal social status, as well as their own trajectories and experiences of socialisation, implies that they often forge complex "in-between" linguistic and cultural identities and senses of belonging. To reflect on these complexities, we analyse multimodal data from transnational- and minority-background children as they participate in an autobiographical activity aimed at promoting linguistically and culturally inclusive pedagogical approaches and participatory action research (PAR). The analysis shows that children's identity constructions fluently intertwine elements from their "home" and "host" languages and cultures with features characteristic of child/youth popular cultures, and with adscriptions to diverse real and imagined communities. These hybrid articulations, which can be described as plurilingual and transcultural, foreground how identity is both an individual and a social process, transversed by different axes, including cultural and ethnic referents, linguistic repertoires, historic, family and personal trajectories, urban cultures and the influence of friends and peers, among others. The identification of these emergent traits in our data foregrounds both the particularities and commonalities of pupils' identity construction, which challenges and reshapes traditional understandings of identity. Finally, this work aims to illustrate how transnational children's complex senses of being and belonging can be recognised and supported through inclusive pedagogical proposals as the one described herein.

**Keywords:** linguistic and cultural identities; plurilingualism; "transculturación"; inclusive pedagogies; transnational background children; participatory action research (PAR); Catalonia; Spain

## 1. Introduction[1]

This article aims to engage with some of the complexities involved in understanding linguistic and cultural identities in a time marked by diversity, transnational mobility and social transformations that question the very meaning of these constructs ([1–3] among others). These circumstances foreground the need to overcome essentialist and reductive notions of culture, language and identity that might result in the invisibilisation or exclusion of certain individuals and communities and move towards more complex and inclusive conceptions that acknowledge the hybrid and fluent nature of current life trajectories, communicative repertoires and senses of being and belonging in our highly diverse societies [4,5].

In this sense, this work aligns with an increasing concern about how "the movement across cultural, linguistic, and (often) geopolitical or regional borders and boundaries" articulates with identity construction and expression "through particular language and

literacy practices across the life span, at home, in diaspora settings (. . .) and in other contexts and circumstances" [1] (p. 57).

As a focal point for this reflection, we analyse ethnographic data from children with transnational and ethnic minority backgrounds in Catalonia, Spain, as they participate in a plurilingual and multimodal arts-based biographical activity named "Travelling suitcases", within the framework of a European project aimed at fostering inclusive educational practices. The particular sociolinguistic and multicultural context where these young people live and study is characterised by the coexistence of national, local and heritage languages and diverse sociocultural backgrounds with unequal social status (see Section 2.1 below). These circumstances, along with their own life trajectories and sociocultural experiences, often marked by transnational mobility, imply that these pupils forge and display complex "in-between" linguistic and cultural identities and senses of belonging. As children navigate through different contexts in their daily lives, these hybrid affiliations and repertoires emerge as either an asset or, oftentimes, as a problematic stance that they might deal with in terms of pride, acceptance, rejection or prejudice, applied both to themselves and to others. Through their engagement in meaningful plurilingual, transcultural and multimodal practices—such as the one described in this work—that allow them to articulate and reflect upon these constructions, children and their adult tutors can create opportunities for reconfiguring and embracing the complexity and uniqueness of current linguistic and cultural identities from a standpoint of richness and respect, and for conceptualising diversity as a constitutive element of our time.

In the following sections, we present the socio-educational setting where this pedagogical and research practice took place, including both the Catalan sociolinguistic context and the specific neighbourhood and school from which our data emerged, as well as the wider European project in which this action is embedded.

We will then introduce the theoretical framework that articulates our analysis around conceptions of linguistic and cultural identity from a psychosocial, sociocultural and post-structuralist perspective [4], as well as on the complementary notions of plurilingualism [6,7], "transculturación" [5,8], "transcultural repositioning" [9] and "in-betweenness" [10–14], all of which contribute to the conceptualisation of linguistic and cultural identities as dynamic, fluid and contingent to the social context and to social relations.

This framework will be then applied in the analysis of data from diverse typologies, guided by qualitative methods, including arts-based approaches, ethnographic observation and fieldnotes, participatory action research (PAR) and semi-structured individual and group interviews of both children and teachers.

After presenting the main results from our analysis, we finally reflect on the potential of such meaningful pedagogical practices and on the role of education in reproducing or contesting and expanding hegemonic understandings of linguistic and cultural identities through plurilingual pedagogies and whole-child approaches that give voice and space to pupils' entire repertoires and life experiences.

## 2. Context

### 2.1. Catalonia, from Bilingual to Pluricultural and Plurilingual

Catalonia, where this research takes place, has experienced intense demographical, cultural and sociolinguistic transformations in recent years. In only a decade, the percentage of foreign residents—in relation to the overall population of the region—has risen from less than 3% in 2000 to more than 15% in 2010, a ratio that has remained quite stable thereafter and is now slightly over 16% [15]. The latest statistics identify the presence of more than 170 nationalities and over 300 languages [16] within the Catalan territory. These demographics have transformed the Catalan sociolinguistic landscape, evolving from its traditional bilingualism—made up of the coexistence of Catalan and Spanish—towards a culturally and linguistically rich and diverse society.

These sociolinguistic transformations have also raised educational challenges, particularly in relation to a well-established bilingual immersion model created in the 1980s,

after Franco's dictatorship (1939–1975), with the aim of protecting and promoting Catalan, the regional minority language that had survived in a state of diglossia [17] along with Spanish, the State's official language, which was (and still is) predominant in many public spaces and in the mass media. After being banned from public life across four decades, Catalan gained a new status as the co-official language in Catalonia and as the vehicular language of instruction and communication in compulsory education, articulated with the instruction of Spanish and foreign languages (usually English), which were taught as language subjects within the curriculum.

While the immersion model has been widely recognized for its role in promoting the local language and creating bilingual Spanish–Catalan citizens, the emergence of a fast-growing plurilingual and pluricultural school population prompted the need to revise its original design to better acknowledge the complex and diverse linguistic practices and repertoires of the Catalan school community [18].

More recently, "The language model of the Catalan education system: Language learning and use in a multilingual and multicultural educational environment" [19] took immersion to a next level by encouraging linguistically and culturally inclusive approaches, methodologies and school practices, such as the articulation of languages and non-linguistic subjects across the curriculum and the consideration of pupils' family languages and cultural backgrounds within school activities. This revised plurilingual immersion model is still in a relatively early phase of development, and many educational institutions and teachers find themselves willing and, at the same time, at a loss on how to bring these inclusive approaches to practice [20]. Indeed, the school context where this research took place can be described as struggling to implement these plurilingual approaches in a linguistically, culturally and ethnically diverse community (see Section 2.2 for a more detailed description).

From a research perspective, all of these transformations have raised significant interest in documenting, analysing and understanding current processes of language and cultural socialisation and identity construction and expression in Catalan educational spaces, including diverse students' perceptions of self and others, and teachers' opportunities and challenges for recognising and supporting these processes [21] (among others). These were precisely the interests behind the participatory action research project from which our data emerged and which are described herein.

### 2.2. Sociocultural and Sociolinguistic Context of this Research: An Institute-School in Catalonia

The school where our analysis took place is located in the province of Barcelona, in the municipality of Montcada i Reixac. It is an Institute-School, which means that since the school year 2019–2020, it comprises preschool, primary and secondary schooling within the same institution, a rare situation, as Catalan public centres usually cover just one of these different stages separately. The aim of this integration is to ensure educational continuity for the students of the neighbourhood from the first year of kindergarten to the end of secondary school.

The school has two lines for each grade, made up of two groups of around 20–25 students each. The school facilities are formed by twenty-six classrooms, where lessons take place, and other rooms for specific activities such as music lessons and science labs.

The site is located in a neighbourhood characterized by the presence of numerous underprivileged families with socioeconomic disadvantages, most of them with transnational migrant backgrounds. According to the latest census [22], the foreign population residing in the district corresponds to 13.7% of the total residents, and the annual growth of migration flows averages 4.18%. The main migrant communities are from African and Asian backgrounds[2].

In line with these sociocultural traits, the centre is characterized by its linguistic and cultural diversity (see Section 2.3, below), with a significant ratio of students from transnational, migrant and minority backgrounds (also including Roma-Gypsies). These circumstances have granted the school with the category of "maximum complexity", a label

conceded by the local administration to those centres that due to their socioeconomic and sociocultural characteristics need extra support and specific measures (e.g., lower ratios of students per classroom or more school staff) to attend to their population.

Despite these specific measures, teachers' reports in focus groups and other, less structured encounters with the authors indicate that there are some difficulties that can be found outside the school, such as stereotypes, prejudices or arguments between different communities, which can have repercussions in the school dynamics. Likewise, teachers also comment on the lack of sufficient staff to deal with difficult situations that might emerge within the classroom. In addition to this, members of the school board have reported significant levels of absenteeism and difficulties in the communication with families, who do not always participate in school activities as teachers would expect [23]. This situation could also be related to the fact that many migrant families have none or very limited knowledge of the official languages of the territory and vehicular languages of the school.

### 2.3. The Sociolinguistic Diversity of the Institute-School and Teachers' Attitudes towards Plurilingual and Transcultural Approaches

To collect the sociolinguistic diversity of pupils, as well as teachers, from the Institute School, questionnaires were administered to the teachers, adapted from those previously carried out in Italy by the project partners from the universities of Bologna (UNIBO) and Turin (UNITO), who were responsible for the first implementation of the pilot actions that are described and analysed later. The questionnaires were adapted to the Catalan context, for example, considering the bilingual policies of the setting. Due to reasons of space, the main results that emerged after the survey are presented and summarized in a schematic way below. The following aspects emerged from the questionnaires:

- There was significant diversity among the profiles of the participating teachers (including preschool and primary school tutors and specialists), especially regarding their previous training on topics related to pluralistic approaches in language teaching and to co-creation.
- The school was already participating in and carrying out projects associated with languages. For this reason, the teachers did not perceive barriers to engage in our project at the level of the linguistic and educational policies of the school centre.
- As far as students' linguistic repertoires were concerned, more than 14 home languages and varieties were spoken both inside and outside the centre, and different varieties of Arabic and of Latin-American Spanish were especially prevalent. Similarly diverse was the degree of knowledge about these repertoires on the part of the teachers, who, in some cases, did not recognise particular varieties and assumed, for example, that all pupils from Morocco had standard Arabic as their home language. In other cases, not all the linguistic repertoires of the pupils were detected; that is, teachers did not always know which languages their pupils knew or used at home.
- Some teachers did not perceive the linguistic diversity of their classrooms as an asset but rather as a barrier to proper communication and learning. For this reason, there were very different opinions among the participating teachers at the beginning of the project. According to some, the activities related to the enhancement of linguistic diversity were perceived as an opportunity, while others framed them as potentially problematic.
- Some critical aspects emerged regarding the use of other languages during lessons, as they could lead to confusion in the pupils. Indeed, there was a tendency on the part of the teachers to associate the use and learning of a particular language with a full-immersion ideology, often described as "One language only" and "One language at a time" [24,25]. These beliefs in language separation became evident in their argument that, for the proper learning of Catalan and other curricular languages, pupils' home languages should be kept out of school.
- The categorisation of pupils' plurilingual practices as problematic might also relate to the fact that 53.9% of the teachers who answered our questionnaire agreed with

the idea that the "ideal" speaker of a language corresponds to a "native" speaker, a referent of language proficiency largely questioned [6,26] and which might allocate less value to the hybrid linguistic repertoires and complex practices of plurilingual pupils[3].

- On the other hand, teachers showed an interest in learning new strategies and plurilingual pedagogical approaches, and, with the implementation of the project, they overcame an initial "fear" to revise their routinary practices and co-create activities with their colleagues, children, families and the university research team.

We now proceed to describe this project in more detail.

### 2.4. The NEW ABC Project

The NEW ABC project, which started in 2020 and will end in 2024, is funded by the European Commission, and it is a Horizon 2020 action. The consortium of partners is led by the University of Bologna-Forlì and comprises thirteen partners from countries of the European Union. Through the collaboration of these countries, nine actions have been carried out through the project and subsequently re-piloted in different areas to test possible ways of adaptability and scalability according to the different sociocultural and sociolinguistic contexts.

The NEW ABC's main objective is the inclusion of children or young people with a migrant background[4], including political refugees, within the societies in which they are welcomed. According to Save the Children [28], very often these minors are not listened to and are not given the opportunity to express themselves or to bring out their skills and abilities. The organisation states that, when children and adolescents are given a voice and become actively involved in their educational process, expressing their skills and desires, this contributes to better social, cultural, educational and linguistic inclusion.

Top-down policies have meant that, for many years, children and young people, especially those with a migrant family background, could not be active agents in their own growth and education process. The NEW ABC project is based on a bottom-up approach, and it aims to improve the agency and empowerment of all possible stakeholders involved, including minors, families, communities and, more generally, society. This ensures the development of horizontal and vertical synergies among all the subjects who are called to participate, promote and co-build together in designing new actions. This new approach could lead to a change that favours linguistic and social inclusion and makes stakeholders the real protagonists of change (https://newabc.eu/, accessed on 5 October 2023).

Due to these reasons, the nine actions of the project are designed to ensure that the academic, social and emotional needs of children and their families are received and listened to in their educational settings and within the host context and society. This implies a holistic view of formal, informal and non-formal education, where decisions are created cooperatively with policymakers too.

The NEW ABC project is founded on four pillars:

- Participatory action research (PAR), which ensures that all the stakeholders' voices are listened to and accepted in different contexts, including also vulnerable or other hard-to-reach participants. A PAR approach also enables the design of new socio-collaborative partnerships and facilitates the social inclusion of vulnerable groups, including children with migrant family backgrounds.
- "Care and compassion", which guarantees a transformative approach enabling all of the community, families and children's needs to be engaged through the project.
- A "whole child" approach, which also includes the "whole school/community" approach. This focuses on the holistic—social, academic and emotional—needs of minors [27] (p. 110) to foster their agency and visibility and to negotiate new policymakers' decisions in the hosting community.
- "Co-creation", which will finally ensure co-designed, co-decided and co-created actions with all participants in order to generate good practices for all the aforementioned stakeholders in and beyond educational settings.

This last pillar is especially important for the qualitative research that we developed as part of our specific action, which consisted of promoting linguistically inclusive pedagogical approaches, and which is described herein. The principles of co-creation have been fundamental to collaboratively constructing activities with teachers and to creating new possible practices and actions within the school context of the Institute-School. Co-creation has also been crucial in giving a voice to a group of children considered more vulnerable, especially those from ethnic minorities and/or migrant family backgrounds, and who belong to communities traditionally less involved in decision making or in explaining to others their social, linguistic and cultural identity.

In this text, we focus on one activity conducted during our repiloting in Catalonia of the action "Teacher training and family involvement in pluralistic approaches to language education", previously conducted in Italy by the Universities of Bologna and Turin. The objective of this action was to create conditions and activities so that pupils' plurilingual repertoires, skills and competences were recognised and valued as assets, resources and opportunities in the school curriculum. With this aim, and through a participatory co-creation approach, we collaborated with the school and families to develop and implement a series of activities that aimed to achieve the following:

- Raise awareness among teachers, students and families about the benefits of plurilingual pedagogical approaches;
- Involve and empower families in their children's education, fostering transversal plurilingual learning and bottom-up synergies, which in turn can support the display and development of their plurilingual repertoires;
- Support teachers in the development of effective practices for the use and appreciation of plurilingual repertoires.

Within the development of the aforementioned action, we focus here on a specific activity called "Travelling suitcases", which was implemented around the celebration of the "mother tongue day" (21 February 2023) to work specifically on issues around linguistic and cultural identity. Before describing this activity and analysing the emerging data in regard to pupils' linguistic and cultural practices and identities, we present here the theoretical framework that is the basis of our analysis.

## 3. Theoretical Framework

To properly describe and analyse pupils' linguistic and cultural identity constructions and expressions as they emerge in the specific activity of the "Travelling suitcases", we need to first elaborate on how identity, and specifically plurilingual and transcultural identity construction and expression, is conceptualized in this work.

### 3.1. Understanding Identity at the Crossroads of Individual, Social and Cultural Factors

Following the multi-theoretical approach to identity proposed by Fisher et al. [4], situated at the intersection of psychosocial, sociocultural and post-structural perspectives, we understand identity construction as a dynamic process that is influenced by individual, social, historical and cultural factors. Such a perspective of identity as a process rather than a fixed condition expands beyond essentialist and reductive notions ascribed mainly to origin, gender, social class and/or ethnicity and towards an understanding of identities as multiple, mediated by social contexts and interactions [29–31] and "constantly becoming" [32] (p. 221).

In line with these principles, a useful definition for identities would involve interactional and situated practices and processes of "organising and reorganising a sense of who [one is] and how [one relates] to the social world" [33] (p. 9), as well as "the social and cultural processes from which one is constructed" [32] (p. 220).

In the case of children and youth, identities are also contingent on pupils' social relations and to their identification with and participation in different communities of practice [34,35]. Consequently, understanding children's identity also implies considering the significant influence of peers and of elements typical of child/youth popular cultures

and of the social media, including their participation or adscription to real or "imagined"—non-physical—communities [36,37]. These physical and/or virtual communities might be as diverse as music or celebrity fan clubs, football and other sports' followers, commercial brands' consumers, YouTube followers or video game players, among others, many of which emerged in our data.

Thus, when focusing particularly on linguistic and cultural identity, issues around language(s) and culture(s) intertwine with these multiple affiliations in ways that connect with the previously described conceptualisation of identity across psychosocial, sociocultural and post-structural perspectives [4]. This expanded, inter-level vision can be extensive to our understanding of both language and culture as influenced by individual and social factors, an understanding that connects to and is framed by plurilingual and transcultural approaches, as explained herein.

### 3.2. Plurilingualism and Plurilingual Identities

Following the definition provided by the Council of Europe's Common European Framework of Reference for Languages [6] and other relevant works on the subject [7,38], among others), plurilingualism refers to speakers' knowledge and use of a wide range of interrelated linguistic and multimodal resources, often in the same interaction, for achieving different goals[5]. This concept has contributed to a theoretical evolution, departing from previous conceptualisations of languages as bounded systems [39], to conceiving language as process and practice, or what some authors have described in terms of "languaging" as social action [40,41]. In this sense, plurilingualism focuses on speakers' entire repertoire of complex and fluid linguistic and multimodal practices, as displayed in situated interactions in a specific time and place.

In terms of linguistic identity, plurilingualism has contributed to challenge the figure of the "native speaker" as the reference and measure to categorize a linguistically competent subject, proposing instead the notion of a plurilingual speaker with an expanding linguistic repertoire made of diverse and uneven skills in the different languages that he/she knows and has had contact with across his/her lifespan. By departing from previous understandings of languages as abstract codes independently from their use in interaction, and of "native speakers" as referents of linguistic competence, plurilingual approaches relate linguistic identities to speakers' diverse and hybrid repertoires; to their agency and resourceful participation in interactional meaning-making processes; and to possibilities of empowerment, creativity, criticality and transformation [42], among others.

Thus, from the standpoint of plurilingualism and plurilingual pedagogies, pupils' display of their entire repertoire, including their home—and often minority—languages or varieties, is acknowledged as a positive and desirable feature for participation, self-affirmation and learning that needs to be recognized, promoted and developed in educational policy and practice [43,44].

From this stance, encouraging the display and development of plurilingual identities in educational spaces would involve allowing and embracing pupils' use of their entire repertoire and hybrid linguistic and cultural practices as resources for identity construction and for learning, promoting awareness of their unique and constantly evolving repertoires and related competences, and encouraging them to reflect on the particularities and value of their own and others' linguistic and cultural diversity.

Indeed, Fisher et al. [4], among others, have signalled the potential of the classroom as a key site "where learners are offered the agency to develop a multilingual identity" (p. 449). Facilitating this process in educational contexts is relevant for two reasons:

> *(a) if learners adopt an identity as a multilingual they may be more likely to invest effort in the learning and maintenance of their languages; and (b) with increasing mobility and greater diversity in communities and classrooms, a multilingual mindset might lead to enhanced social cohesion in the school and beyond.* (p. 449)

This identity construction, the authors claim, should be an explicit and participatory process where pupils are given opportunities to identify with specific languages or varieties

as part of their linguistic repertoire, to consider and (re)conceptualise their identities as linguistically and culturally diverse and to become aware of the possibilities of transformation and change involved in these identifications.

Given that these approaches allocate a significant responsibility to schools and educators in facilitating pupils' identity development, it seems relevant to further explore how to better comprehend and legitimize children's repertoires and practices, especially when these transgress and expand traditional understandings of languages and cultures as fixed sets of features.

To properly account for the fluid nature of cultural identity processes and of senses of affiliation and belonging as they emerge in our data, we draw here on other, complementary constructs that contribute to this perspective from the specific standpoint of ethnic minority and transnational background children: the notions of "transculturación" [5,8], "transcultural repositioning" [9] and "in-betweenness" [10–14].

### 3.3. Cultural Identity Construction Regarding Minority/Transnational Background Children: *"Transculturación"*

The concept of "transculturación" was first coined by Ortiz [8] in opposition to rigid conceptualisations of cultural contact in terms of acculturation and assimilation to foreground the interactional, dynamic and transformative nature of cultural encounters, as they give birth to new cultural understandings and identities.

More recently, García, Homonoff-Woodley, Flores and Chu [5] expanded on the original concept (maintaining its original name in Spanish) to refer to minority- and transnational-background children's ability to "straddle across cultures and to perform features of what might be considered different 'national [and, we would add, ethnic and other] cultures' as their very own in interaction with others" (p. 812).

This definition emphasises pupil's ability to move past difference and to challenge and destabilise socially built oppositional dichotomies around language and cultural identities such as home/host, native/foreign, local/newcomer and similar exclusionary constructions around nativeness, belonging and place. "Transculturación", then, refers to building "a common collaborative 'in-between' space that transcends linguistic and cultural differences", articulating diverse and intertwined subjectivities and worldviews and displaying hybrid cultural backgrounds, experiences and senses of being and belonging (opcit, p. 799).

García and her colleagues also highlight the role of educational contexts and agents in supporting the development of pupils' fluid, transcultural identities, which would boost them "to perform academically and socially in ways that help them grow beyond the static positions to which they are often relegated" (p. 817).

A similarly relevant concept in relation to framing pupils' hybrid linguistic and cultural practices in terms of competence is that of "transcultural repositioning". The notion was developed by Guerra [9] to account for transnational-background children and youth's ability to flexibly adapt their linguistic and cultural practices in order to navigate the diverse sociocultural situations in which they participate within and beyond school settings. The author claims the relevance of such transcultural repositioning to understanding "how identities and cultures can be hybrid and transcultural, even as they are situated in and constructed through local and particular practices" (p. 138).

Other authors (e.g., [10–14]) have also explored issues around identity construction and senses of belonging in youth with migrant backgrounds, more specifically in the case of communities from Latin-American origins in the USA and Europe. These studies emphasize the negotiated and polyhedral nature of these pupils' identities, which they describe in terms of "complex transnational belongings" and "in-betweenness". One of the first authors to explore these phenomena was Anzaldúa [10], who used the term "Nepantla", a Nahuatl word, to refer to the space in between two worlds, where "you are not this or that" but actually "seeing from two or more perspectives simultaneously" (p. 276).

When applied to youth and children's identity construction, this multiple positioning implies the following:

> *For many of the youth, their identities were forged in the in-between spaces of here and there, origin and destination, as they negotiated and straddled multiple worlds—the state, family, peers, school, and popular culture. Therefore, these youth of immigration asserted expanded notions of belonging as they enacted hybrid, "flexible," and "polycultural" citizenships* [14] (p. 84).

All of the above serves to articulate our own understanding of linguistic and cultural identity construction, as well as to properly account for the complex profiles of our young participants. So far, we have referred to most of the pupils that took part in this project as transnational-background children, a label that will be briefly elaborated herein.

Our understanding of transnational children and youth draws on the notions and ideas described above, as well as on work by Duff [1], who proposes an expanded conceptualisation of the term beyond its original focus on physical mobility:

> *Whereas studies might once have focused mainly on first-generation adult immigrants' transnationalism, there is now a growing emphasis on the mobility of children and youth as well; on virtual and psychological connectedness (and not just physical mobility and interactions); and on multigenerational experiences affecting languages, individuals, and communities in transnational spaces* (p. 57).

Thus, following Duff and others, transnational-background children in this work, most of whom were born in Catalonia to "first-generation" migrant parents, construct and display complex identities and engage in social relationships and cultural processes that articulate many different axes and experiences across physical, as well as virtual, spaces.

Duff, as well as most of the authors/works described in the previous lines, also emphasizes the role and responsibility of educational institutions and advocates for creating "safe spaces" in school settings where these multiple, transnational and expanded identities are acknowledged and supported through sensitive and inclusive pedagogical approaches.

Finally, we would add, pedagogical initiatives that aim at inclusiveness and identity affirmation should also be open to acknowledge and critically reflect upon pupils' interests, preferences and practices in relation to features from child/youth popular cultures, including issues around consumerism [45], social media and their sense of membership or aspirations of belonging to multiple real or imagined communities ([36,37] see above).

In general, the previously described theoretical and analytical approaches that foreground the construction of transcultural identities that do not operate on an exclusionary axis around being and belonging, and which can legitimate, expand and/or transform participants' understandings of themselves and of others, clearly align with our research and pedagogical stance. Indeed, the collaborative plurilingual/transcultural activity that will be described and analysed in the following sections, named "Travelling suitcases", aimed to open spaces for children's hybrid, complex, "straddling" linguistic and cultural repertoires and identities to emerge, be reflected upon and be recognised by their school community. We now proceed to explain the methodological aspects of this research, including the wider project in which it is embedded and the specific activity that gave was to our data.

## 4. Methods and Materials: Analysis of the "Travelling Suitcases"

The methodology is qualitative and based on an ethnographic and emic approach [46]. Field notes, on-site observations and conversations with teachers and children, questionnaires for teachers, semi-structured interviews with teachers and children, and focus groups with teachers were employed throughout the research. Data were also collected using "postcards" at the end of the activities to gather children's opinions and preferences and to find out what they had shared at home about the experience.

At the beginning of the project, our research team met with the schoolteachers to present the basic principles and objectives of NEW ABC, focusing especially on PAR and

co-creation. Firstly, some examples of activities to promote plurilingual pedagogies were presented as a model for their consideration on how to integrate them into the school curriculum) [23]. Secondly, a period of intense ethnographic observation was carried out in the classrooms to collect information regarding the school's working methodology and to become aware of how to integrate the project objectives into the work that the teachers were carrying out. Subsequently, data were collected from the pupils themselves regarding their linguistic repertoires through two well-known autobiographical activities: the "flower of languages" [47] and "linguistic biographies" [48,49]. These activities consisted of the creation of either a flower or a silhouette representing children's linguistic repertoires, as well as their trajectories and everyday practices in regard to languages. Different groups adapted the activity to be developed either collectively or individually, and the final outcomes were displayed in the school corridors, marking a first step in the recognition and visibilisation of the school's diversity.

Finally, after this first phase of observation and data collection, different plurilingual, multimodal and transcultural activities were co-designed with the teachers. Regarding our role as researchers, we were allowed to observe and support or lead all the activities, depending on their complexity and the confidence of the teachers engaged. One of these activities was the "Travelling suitcases".

The idea of the "Travelling suitcases" emerged from both a concern with opening spaces for children's home languages and out-of-school practices to emerge within school hours, and with creating opportunities for pupils to express their plurilingual and transcultural identities in creative ways and in their own terms, giving them voice and recognition. The idea stems also from the desire, shared by researchers and teachers, to explore the linguistic, social, relational and transcultural elements that conform pupils' identities and to enhance their home languages and diverse repertoires on the framework of the "mother tongue day", which is commemorated on the 21 February.

The activity involved producing an artistic collage representing a suitcase rich with images of objects, words, concepts, pictures, drawings and other expressions representing those things that were more important to pupils' self-perceptions and lives and that they would like to take with them either from home to school, or in a hypothetical travel or moving abroad (e.g., from their country of origin to another place, or from Catalonia to other parts of the world). The suitcases were created by the children from cardboard folders to which twine was glued and used as a handle. Each suitcase could then be filled freely, both on their outside and inside spaces. While the instructions were quite open and flexible, they stated that each pupil should include in his/her suitcase at least two words/concepts from their home/heritage language(s), or any other important language for them, expressing important concepts that they would always want to keep with them in the event of travel or displacement. Parents and families were also involved at home, deciding together with their children on which photos or other graphic elements to include, and in choosing and writing the words in their different languages.

The "Travelling suitcases" activity was an arts-based multimodal production. To conduct the activity, we also employed visual methods of social research that, according to Rose [50] (p. 24), are inextricably linked to contemporary visual culture and society itself:

> *Imbued with the everyday and conversations, meanings and affects, images [. . .] are made, shared, shown in multiple ways. They can be representations that convey culturally relevant meanings; they can be tools for thinking; they can evoke the ineffable [. . .].*

Moreover, Stocchetti and Kukkonen in Frisina [51] reaffirm the importance of using images in social research. Indeed, the authors claim that images are never objects to be analysed in an autonomous and isolated manner but must be contextualised within social practices in which they obtain meaning.

To begin the multimodal activity, the first step consisted of the creation and presentation, by the researchers/authors of this text, of our own "Travelling suitcases", in line with our role as active participatory observers [46] and our epistemological position as transnational, transcultural and plurilingual subjects within the sociocultural Catalan con-

text. We filled each of our suitcases with words that were meaningful to us in our home languages, cutouts of labels or containers of favourite products from our original, current or other settings. We also drew our transnational trajectories, from our countries of origin to Catalonia, indicating the most significant places and most relevant experiences. In addition, we included verses of songs, extracts from novels or poems, painted our favourite colours or symbols, and included proverbs or sayings. Through this exercise, we modelled the activity in terms of our own personal and unique transnational trajectories, repertoires, experiences and senses of being and belonging, while also considering and including sociocultural preferences and affiliations that might be shared with other members of our generation and/or with the different collectives in which we take part.

After this, we invited children to create their own suitcases in collaboration with their families and close ones. In total, around 140 students from 6 different primary school groups were involved (2 second-grade classes, with children aged 7-to-8 years old; 2 fourth-grade classes, with children aged 9-to-10 years old; and 2 fifth-grade classes, with children aged 10-to-11 years old). For each class, two teachers and the educational coordinator also participated, making a total of seven teachers. Once created, the suitcases were presented by the children to their classmates in their preferred languages. Any word or explanation in a home language or variety not known to all classmates was explained by the pupils themselves. Thus, the activity opened spaces to render pupils' family languages visible and to discuss differences and similarities between languages and varieties. It also enabled pupils to express identities and affiliations (for example, to the Gipsy community, as well as to other real and/or imagined communities from popular youth cultures).

Subsequently, interviews were conducted with children from each participating class who wished to explain their multimodal productions in detail to the researchers[6].

In total, 50 interviews of approximately 10 min each were conducted on the 21st, 23rd and 29th of March 2023. The interviews followed a semi-structured question guide with opportunities for the researchers and children to expand on the topics, ask extra questions or recount other aspects that they considered important. The original, basic guide included the following questions:

- Can you explain your suitcase to us?
- Did you enjoy this activity? Why?
- How did you feel about using your home language(s) at school?
- What have you liked most about the whole project so far?
- What would you like to do now?
- Is there anything else you would like to say or ask?

The questions were asked either in Spanish or Catalan, but the answers could be given by the children in their preferred languages or varieties, or in plurilingual modes. Most of them answered in Spanish, some in Catalan, and some in plurilingual modes including both languages; one child did it in Arabic and another in English, also combining features from their home and school languages. The recordings and notes made from the interviews were then reviewed by the researchers, and the most crucial and salient points in relation to the study of pupils' plurilingual and transcultural identities were selected and transcribed.

At the end of the whole process, two focus groups of about one hour each were conducted, one with preschool teachers and the other with primary-school teachers. For reasons of space and subject matter, we focus only on the answers given by the eight primary-school teachers in relation to the "Travelling suitcases" activity in the next sections, as pupils attending preschool conducted different activities on the "mother tongue day".

Finally, at the end of the course, the children were asked to fill in a postcard indicating if and how much they liked the project, what they enjoyed the most about it and what they shared at home with their families. In the upcoming data analysis and discussion of the results, we present a few examples of children's reflections on the activity.

Below, we present examples and the analysis of the data mentioned above:

- A sample of the "Travelling suitcases" created by the pupils, commenting on the salient features of their multimodal creations;

- A selection of fragments from the interviews with the children who participated in the activity;
- Extracts from the focus group dedicated to the primary teachers;
- Fragments from the postcards completed by the pupils at the end of the course.

For each type of data, an initial and transversal thematical analysis of the entire corpus was conducted. Thanks to this, it was possible to identify and codify the main emergent themes [52].

## 5. Results

The identification of themes emerged from a qualitative analysis based on ethnographic and emic approaches; that is, that they emerged in a salient and systematic way from our multimodal data, rather than as pre-established categories of analysis. Thus, the in-depth observation of multimodal data made it possible to identify and group themes into six salient axes, which aim to describe the complex and intertwined nature of pupils' identity construction in this particular activity. These axes are as follows:

- Identity as hybrid and transcultural;
- Identity and the revindication of ethnic minority and/or religious backgrounds;
- Identity and features from popular child/youth cultures;
- Identity as personal and family history and trajectory;
- Identity as plurilingual and linguistically inclusive;
- Identity as social, relational and interactional.

These axes are not perceived as immobile, bounded and fixed categories; instead, they represent a possible way of organising the emergent information and offer us the possibility of structuring and analysing a complex phenomenon with clarity and in depth. Some of the proposed axes intertwine and may, in fact, overlap with each other and can be subjectively reinterpreted by other researchers or participants in the research. They are, therefore, in no way intended to provide a close or unique view of the dynamic phenomenon we have observed, namely the construction of plurilingual and transcultural identities.

The aforementioned axes emerged from the observation of co-constructed multimodal products in which we as researchers also participated. In this regard, according to Manzo [53] (p. 110), "collaborative research can greatly enhance ethnographic technique and analysis". Indeed, Burgois and Schonberg [54] (p. 192) state the following:

> *While participant observation is by definition a subjective method that requires systematic self-reflection, within a coproduced ethnographic process, collaborators have the advantage of being able to constantly share and examine interpretations and insights.*

Following Manzo [53] (p. 193), within a collaborative framework in which activities are co-constructed not only between researchers and teachers but also with pupils, "the potential of an applied ethnography aims both at promoting emancipation and self-determination of research subjects and at creating a critical and public debate"—in our case, a socio-educational debate. To deepen the themes that emerged from our initial observations, we decided to complement the multimodal data with interviews with pupils, focus groups with teachers and postcards filled in by the children.

It is necessary to point out that the activity undertaken by the pupils took place three months after the end of the football World Cup. In numerous "Travelling suitcases", the children placed features related to this sport that could have been influenced by this event. Similarly, we noticed that many of the aspects that children focused on matched those chosen by the researchers, so that the presentation of our "Travelling suitcases" might have shaped their creations accordingly. We would like to highlight this as a possible limitation of the research, noting it as a suggestion for future replicability. Furthermore, we noticed that, within each class group, children mutually influenced each other, as certain elements were included in the suitcases of almost all participants. Finally, pupils' selected features also varied according to their author's age. While children from second grade focused more on favourite foods and pictures from their childhood, fifth graders tended to include



images of footballers, song lyrics and clothing brands. This influence among peers was also visible in their choice of relevant words in their home or favourite language(s).

*5.1. Children's Productions: Examples of Multimodal Creations*

Three examples of suitcases co-constructed by pupils with support from their teachers and families are shown and described below, expanded by pupils' explanations of their creations.

The first example is from Nela[7], an 11-year-old girl of Ghanaian origin who moved to Spain with her family (as the first two flags indicate in Figure 1 below). In her suitcase, Nela pasted two important words for her, namely "friendship" and "family", arguing that "family is what gives us happiness and everything else" and that "if you want to say something to someone, and you are at school, and you have a friend, you can express yourself"[8]. Nela also pasted a picture depicting her favourite food, "fufu", a meat dish cooked in Ghana, along with marshmallows. Her suitcase also features lyrics from her favourite song, which was by Shakira, and symbols of the Ghanaian football team. Nela also depicted France, a country she would like to visit, alongside Ghana and Spain. Her favourite name, Michelle, appears along her favourite brands, Lefties, Primark and Zara. She would like to try Starbucks; her favourite colour is pink, and she likes to play the videogame Roblox online because "you can play in all languages". Concerning Ghana, what she likes most is that her grandmother lives there and how people behave.

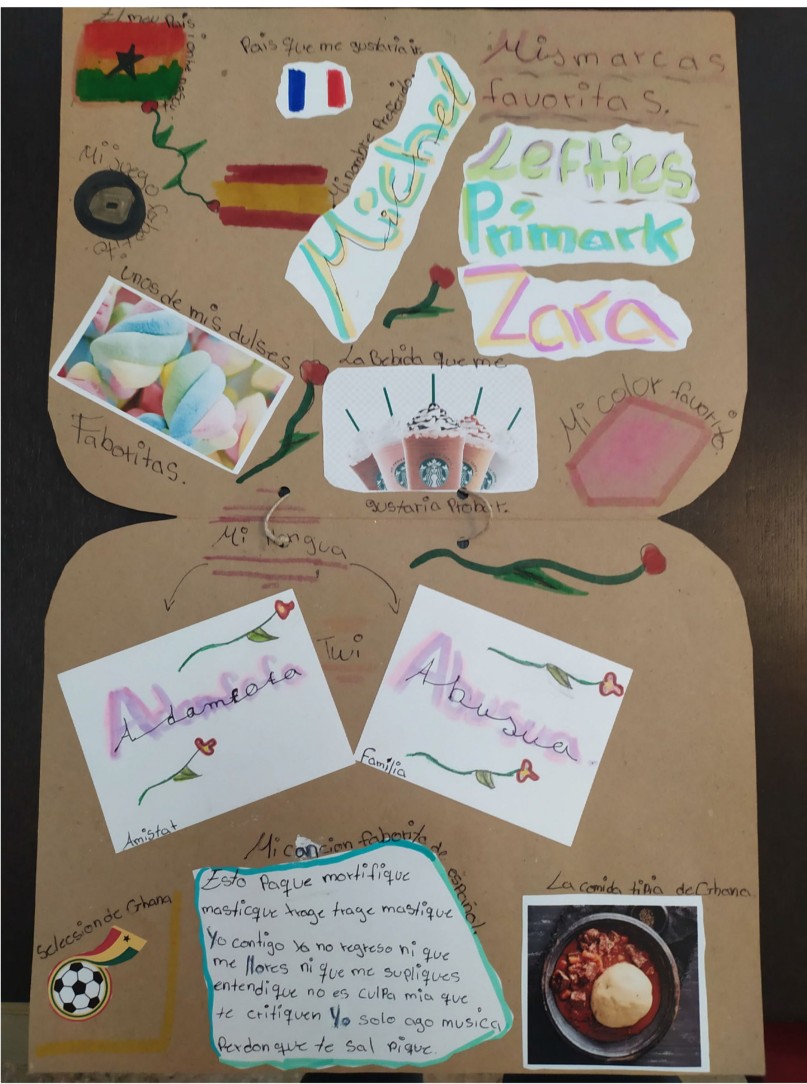

**Figure 1.** First example of a child's "Travelling suitcase": Nela.

A second multimodal product was created by Rani, a 10-year-old girl of Moroccan origin whose home language is Arabian Riff. As we can see in Figure 2 below, Rani incorporated the Moroccan flag in her suitcase and a picture of couscous, explaining that what she likes best about the country is the cuisine. Like Nela, Rani also included the symbol of her favourite football team, Morocco. As for her choice of relevant words, she listed "family" and "friendship" with help from her mother, as she can read but not write in Arabic. In her "Travelling suitcase", we can also find cities from Morocco and Catalonia that she has visited, her favourite colour and some lyrics by Shakira.

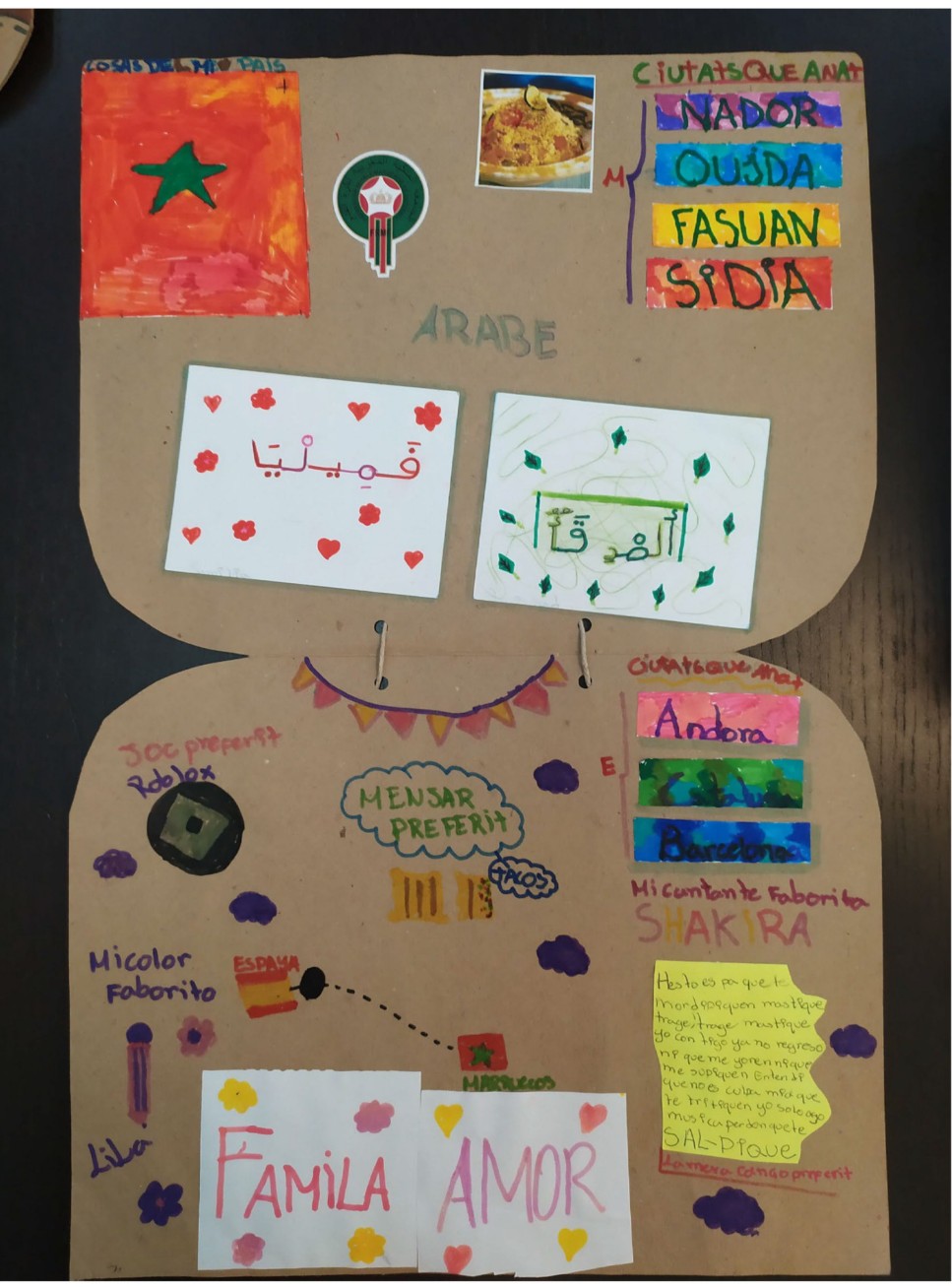

**Figure 2.** Second example of a child's "Travelling suitcase": Rani.

The last example is the "Travelling suitcase" of Dana[9], an 11-year-old girl born in Paraguay whose home languages are Guaraní and Spanish. In her creation, (see Figure 3 below), she drew the Paraguayan flag, clarifying that although it looks very much like the French one, "actually it is not". She also pasted pictures of her favourite football players;

some verses of Shakira's latest song; and her favourite foods—marshmallows and pizza with cheese. Furthermore, the two words she chose were "water" and "happiness", the former because it is the shortest word in Guaraní, spelt as "Y"; and the latter because "we all have to be happy". Also included in her suitcase are her favourite colour; her favourite food chain, Viena; and the game she likes best, Roblox, which Dana likes to play in English. The girls explains that the activity was co-constructed with her mother from a distance, as she was in Paraguay at the time.

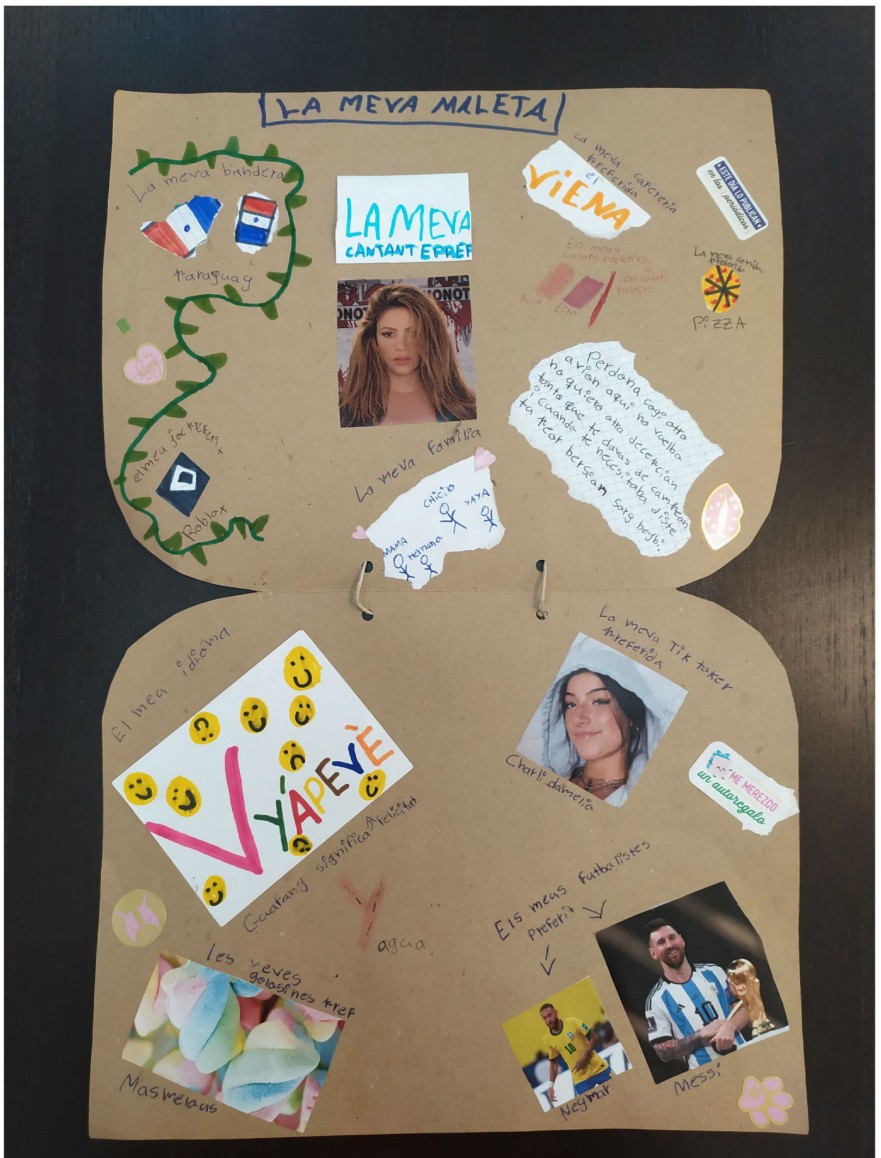

**Figure 3.** Third example of a child's "Travelling suitcase": Dana.

*5.2. Children's Perceptions*

As mentioned in the previous sections, the children filled in a postcard at the end of the course which allowed us to gather their opinions about this and other activities from the project. We include here some descriptions and quotes taken from these productions.

In Nela's postcard, we find an explanation of what she liked best about the suitcase creation. According to her, through this activity, she was able to do the following:

*I was able to share things from my country with the whole class. Moreover, I enjoyed doing the interview" [where she explained her suitcase to us]. At home, she explained that "at school, we made a suitcase about Ghanaian things, and I explained to my mum everything we did, and I showed her the words, and my mum helped me write them down.*

Another girl from second grade wrote in her postcard that "I liked making the suitcases because you put things from your family, and it is like travelling". At home, she described the moment when she explained her suitcase to her classmates with a friend and how they had laughed a lot.

A third child wrote that she really enjoyed decorating and explaining her suitcase. At home, she explained that she had chosen important words in her language and that it had been very crucial for her. She also recounted that the activity had been carried out on "mother tongue day" as a relevant fact.

Children's voice and impressions about this activity were also collected through semi-structured interviews, which included the following information:

Some children, including Nela, highlighted the importance of using their home language(s) at school, as seen in the following quote: "I would love to learn new languages. I thought it was good to use the home language at school because there are some people who don't know it and I can explain it to them, so they will understand it better".

This also emerged in Rani's interview: "It would be important for other people to know what we speak at home and understand a bit more of our language". She also commented that she would like to have opportunities to talk about traditions and compare them with those of her classmates.

On a different note, Dana expressed that she had not particularly enjoyed the activity because she speaks Guaraní at home with her grandparents but cannot write or use it in other contexts. Still, she conceived that "I would like to play games in Guarani and in other languages that are not known".

*5.3. Teachers' Perceptions*

As emerged from focus groups and informal conversations, many teachers were surprised by the interest shown by classmates during their peers' suitcase presentations. Teachers were also surprised at children's ability to understand each other and to comprehend different languages heard for the first time, commenting on the affordances of intercomprehension between similar words in different languages. One teacher reported how, during the explanation of some selected words in Urdu, children realised that the words were written in Urdu but pronounced in English. They were, in fact, loans from English written in the Urdu alphabet. The children asked their peers about this, intrigued by these coincidences and differences. In addition, teachers noticed how pupils' initial attitude of shame when reading words in their home languages or showing pictures changed as soon as the whole class became interested, promoting transcultural dialogue and exchange.

Teachers foreground that the opportunity to give voice to different languages and cultural features changed the classroom patterns of participation. Children who usually intervened less and had limited proficiency in Catalan were able to participate proudly and to share aspects of their home language and family cultural backgrounds. Referring to the school policy of encouraging children's use of Catalan and placing explicit limits on Spanish and other home languages, one teacher expressed that "maybe if we are telling them that Catalan is spoken here, but your language is equally important as any other, even if the vehicular language is [Catalan], there is an openness of mind". Supporting this argument, another colleague stated that, through this activity, children "became protagonists within the classroom [...] For example, in the case of Guaraní, one child was the only one who knew and explained things in that language".

## 6. Discussion

After a thorough analysis of the pupils' "Travelling suitcases" and other related data, a sample of which has already been presented above, we were able to identify a series of six

salient axes emerging from our data. In this section, we intend to delve further into these different axes, situated at the crossroads of pupils' identity construction and of linguistic and cultural identity expression in particular. While separated here for analytical reasons, these proposed categories intertwine; may overlap; and are open to further elaboration, revision and transformation.

### 6.1. Identity as Hybrid and Transcultural

The first axe refers to hybridity as a relevant feature emergent in all of pupils' identity expressions. Categorising identities as hybrid entails, on one hand, the fact that pupils fluently combine features that might be associated with different "home" or "national" cultures as their own, as seen in the examples of national flags (Morocco and Spain) or favourite foods (fufu and marshmallows), among many other. Hybridity also involves "mixing" very different cultural referents in their multimodal creations, as seen when pupils displayed national symbols and features of child/youth popular cultures (e.g., video games) side by side within the same production. In sum, both "culture" and "identity" emerge as open and intersectional categories that can encompass a variety of different referents in non-exclusionary ways.

### 6.2. Identity and the Revindication of Ethnic Minority and/or Religious Backgrounds

A second significant axis regarding identity that emerged from our data has to do with children's sense of belonging to a specific community with its own history and traditions, as seen in the display and revindication of their ethnic minority roots. This axis was especially salient—but not exclusive—in the case of children from Roma (gipsy) family backgrounds, who were eager to include in their suitcases and explain to researchers, teachers and peers the symbols, traditions and values of their community. These revindications also involved, in some cases, references to religious traditions and values, as in the case of some Muslim-background children who included and explained the relevant meaning of Ramadan in their lives[10]. A further important aspect that emerged during the interviews regarding Ramadan concerns the identity construction of several children who, despite having different cultural backgrounds, identified with a sense of belonging to the same religious community. One Pakistani girl, in fact said, that "we Pakistanis with Ramadan are the same as Moroccans".

### 6.3. Identity and Features from Popular Child/Youth Cultures

This axis might be one of the most salient, as it systematically emerged in pupils' creations in terms of pop singers and lyrics, football teams and players, fast food chains, social media apps, computer games and brands of clothes, among other elements that we might identify with urban popular cultures that transcend national, social and other "borders" among communities. From a critical perspective, several authors have noticed the link of these features to globalization, consumerism, materialism and neoliberal ideologies [45,55,56], as well as their dominant Western/mainstream origins, which should be reflected upon with children. While different from the type of engagement one might experience with a "real", physical community (as in ethnic minority groups and religious congregations), these affiliations to virtual or "imagined" communities [36,37] or what popular culture would describe as "followers", clearly play a relevant role in pupils' identity constructions to the extent that all of the child participants included some of these features in their creations.

### 6.4. Identity as Personal and Family History and Trajectory

This axis emerged in children's productions in several multimodal ways, for instance, as pictures of them as babies or younger kids; family sites that were dear to them (e.g., mountains, villages and grandparents' homes); favourite places visited; and "timelines" describing their life trajectories, including their families' origins, host country (Catalonia/Spain) and other significant geographical sites. The role of life trajectories in children's

identity constructions talks about their revindication as transnational and transcultural citizens and problematises reducing identities and senses of belonging to essentialist notions such as place of origin or nationality in current times.

### 6.5. Identity as Plurilingual and Linguistically Inclusive

Plurilingualism imprints children's identity productions in several ways. Their "Travelling suitcases" are essentially plurilingual, as they include texts in several languages, along with other multimodal forms of expression, within the same message. Furthermore, by including their home languages—which were traditionally banned from school practices—in their suitcases and presentations, their plurilingual practices challenged and transcended traditional boundaries and ideologies about language separation, in line with the principles of plurilingual approaches. By displaying their entire linguistic repertoire in free and creative ways, children were able to visibilise and value their home languages and mixed linguistic practices as useful and valid resources for educational purposes, side by side with the local and school vehicular languages.

### 6.6. Identity as Social, Relational and Interactional

Finally, our data foreground the essentially social and situated nature of identity construction, strongly contingent on the specific context and children's relations and interactions with others as members of diverse communities of practice, including the classroom. This becomes evident in the many coincidences found in pupils' creations, including similar categories and, within these, the selection of common elements (specific words, songs, games, etc.). These commonalities might be attributed to the model provided by researchers, to pupils' sharing of their ongoing creations and to specific trends and contextual factors (the World Cup, Shakira's recently launched song, etc.), among other circumstances. This relational factor also refers to families' involvement and collaboration in pupils' creations. Consequently, understanding children's identity expressions implies considering the significant influence of peers and other actors (researchers, teachers, parents, and others) with whom the process is developed, as identity construction also implies "being part of" wider processes "with" others. This dimension clearly embeds some of the previous axes in relation to pupils' affiliations with other physical and virtual communities.

### 7. Conclusions

In this contribution, we analysed how the fluent linguistic and cultural identities of transnational, plurilingual pupils emerge in their multimodal productions as dynamically shaped by personal, sociocultural and interactional factors. As stated by Fisher et al. [4] and taken up in our theoretical framework (see Section 3.2), identity construction is an ongoing process at the intersection of psychosocial, sociocultural and post-structural perspectives in which children express, negotiate, transform and (re)conceptualise the multiple features and circumstances that conform their complex and plural senses of being and belonging in interaction with others.

Moreover, in the case of children from ethnic-minority backgrounds and/or transnational family trajectories, our analysis supports previous works that show how this process of identity construction involves the ability to "straddle" across different linguistic and cultural references, building unique hybrid "in-between" transcultural repertoires and multiple non-exclusionary affiliations. As their multimodal creations show, pupils articulated expanded identities that move past, challenge and destabilise traditional linguistic and cultural boundaries. These assemblages problematise dichotomic and exclusionary categorisations of people and communities that have defined identities and belonging around fixed elements such as naiveness or origin for too long. In this regard, Cohen-Emerique [57] has stressed the importance of avoiding simplified, reductive and univocal representations of otherness, since identity is constantly evolving, as are the changes to which it is subject in interpersonal and transcultural relationships. "Reducing identity means reducing it to a univocal and stereotyped dimension that does not belong to it. It means schematising it,

classifying it, simplifying it and generalising it" (Camilleri 1980 in Cohen-Emerique [57] (p. 35).

To move past reductionism and simplification when approaching linguistic and cultural identities, we propose here a series of axes, emergent from our data, which might contribute to identify and reflect upon the many and diverse features involved in transnational pupils' senses of being and belonging. These axes intend to account for the dynamic social, cultural, historical, community and global practices and influences in which pupils are immersed, although they are not to be considered univocal or fixed but intersectional and open to further elaboration (see Section 5).

In the previous sections, we also reflected on and tried to illustrate how educational settings can open spaces for the emergence and valorisation of pupils' multiple identities, repertoires and affiliations through the implementation of plurilingual and transcultural activities. Such inclusive pedagogical approaches are also crucial in making teachers and pupils aware of their affordances for personal and educational development and for promoting awareness, empathy, dialogue, empowerment and critical reflection.

In this sense, through the activity of the "Travelling suitcases", pupils were given a voice to foreground their multiple, hybrid, plurilingual and transcultural identities, repertoires and affiliations; to reflect and create definitions of themselves in their own terms; and to open a dialogue with their peers, teachers and families. These affordances are in line with the pillars of the NEW ABC project, based on a holistic view of pupils' academic and personal development ("whole-child approach"), as well as in participation and co-creation. Hopefully, these inclusive pedagogical principles will transform previous school dispositions regarding children's hybrid repertoires and practices, encouraging teachers to conceptualise them as assets and to promote their development in curricular activities. In this sense, we take up the indications from García et al. [5], who emphasise the importance of education in supporting the development of fluid identities that can, in turn, help to broaden and enhance academic skills, including those particular to transnational and plurilingual children.

To support our argument, we share here an excerpt from an interview including one researcher (R) and two girls from Arabic backgrounds (C1 and C2) regarding the use of their linguistic repertoires as a vehicle for learning the school's vehicular language (Catalan).

R: Did you like doing the activity in different languages?

C1: Yes, because it makes me feel happier. I liked using Arabic a lot, so other people know what Arabic is and it's something that I like from my heart.

C2: I thought it was good to use the home language in this activity because it reminds me of my family. Using the home language is good because you learn more. I would like to use the home language more in class.

C1: I thought it was good to use my language here at school because I liked speaking Arabic. It's in my head when I use Arabic at school.

R: Ah, it's in your head, but then you speak Spanish or Catalan in the classroom?

C1: Yes.

R: Was it the first time that you were able to use Arabic at school?

C1: Yes.

R: And how did you feel at first?

C1: Embarrassed.

R: And then this feeling changed a little bit?

C1: Yes, I am very happy.

R: And would you like to do it more?

C1: Yes

R: And at what times? At playground time, when you play with your friends, or during classes?

C1: During classes.

R: And why do you use Arabic in your head, does it help you?

C1: It helps me, and I remember how to speak Arabic.

R: In which ways using Arabic in your head helps you?
C1: I think in my head in Arabic and then I say it in Catalan.

The fragment describes the girls' strategic and competent use of their linguistic repertoire for the development of both their home and school languages, an action that Guerra [9] defines as "transcultural repositioning", that is, skillfully moving back and forth between different languages and adapting one's linguistic practices to navigate diverse situations. This navigation also involves emotional processes, described by the girls in terms of shame at first and eventually of satisfaction and happiness.

Indeed, the positive emotional affordances of working with identities through pedagogical actions are at the core of our work, as they systematically emerged in children's accounts and multimodal constructions. To further illustrate this point, we share here a fragment by a 7-year-old girl who, after explaining different features from her suitcase that coincided with those from her peers, described her particular pride in celebrating Ramadan:

> *Today I am wearing a yellow T-shirt because I didn't have a gold one, and it is for the Ramadan celebration that has started. Today is a special day, we say Ramadan Mubarak, we make henna, we eat with the moon and the stars, it is like staying next to them, and we receive money at home.*

We close these lines by embracing the emergence of such accounts of relevant life events in terms of "affirmative action", as they allow minority-background pupils to bring their particular cultural experiences to school and to reposition themselves and their communities in terms of pride and legitimacy.

**Author Contributions:** Conceptualization, C.V.R. and V.T.; Introduction, C.V.R.; Context, V.T.; Theoretical Framework, C.V.R.; Methods and Materials: Analysis of the "Travelling Suitcases", V.T.; Results, V.T.; Discussion, C.V.R.; Conclusions, V.T. and C.V.R. All authors have read and agreed to the published version of the manuscript.

**Funding:** The research has been conducted as part of the NEW ABC which has received funding from the European Union's Horizon 2020 research and innovation program under grant agreement N°101004640. This publication reflects only the authors' view. It does not represent the view of the European Commission and the European Commission is not responsible for any use that may be made of the information it contains.

**Institutional Review Board Statement:** The Study was conducted in accordance with ethical procedures by the European Commission, as well as by the Research Ethics Committee of the Autonomous University of Barcelona (UAB). The UAB Ethics Committee on Animal and Human Experimentation (CEEAH), which met on 28-05-2021, agreed to report favorably on the project with reference number CEEAH 5568, entitled "NEW ABC: Networking the Educational World".

**Informed Consent Statement:** Written informed consent was obtained from all subjects involved in the study.

**Data Availability Statement:** More data are available at the following links: https://newabc.eu/ and https://webs.uab.cat/miras/en/ (accessed on 5 October 2023).

**Conflicts of Interest:** The authors declare no conflict of interest.

## Notes

1. This text has been created within the framework of the New Abc project. New Abc has received funding from the European Union's Horizon 2020 research and innovation program under grant agreement N° 101004640. This publication reflects only the authors' views. It does not represent the view of the European Commission and the European Commission is not responsible for any use that may be made of the information it contains.

2. For more information, visit the official page of the Statistical Institute of Catalonia at the following link: https://www.idescat.cat/emex/?id=081252&lang=en (accessed on 5 October 2023).

3. The idea of the native speaker as the referent for language proficiency has been extensively questioned, among others, by the Council of Europe's Compendium of the Common European Framework of Reference for Languages (Council of Euroe [26]). The framework emphasises that the figure of the native speaker should be abandoned as the sole and defining model of linguistic

competence, particularly for the C2 level. The new model proposed by the CEFR envisages "a 'mastery' [...] in the degree of precision, appropriateness and ease with the language which typifies the speech of those who have been highly successful learners" [26] (p. 38). Furthermore, the abandonment of the native-speaker model opens up the possibility of a plurilingual and pluricultural view of language learners as social actors within different sociocultural contexts. Plurilingual and pluricultural competencies are considered in the CEFR as a repertoire of competencies in different languages or language varieties. It, therefore, becomes the task of language education to develop the linguistic and intercultural repertoires of each individual language learner. For further information, visit the following webpage: https://www.coe.int/en/web/common-european-framework-reference-languages/level-descriptions (accessed on 5 October 2023).

4   For children with a migrant background, we refer to those defined by Eurydice as follows: "newly arrived/first generation, second generation or returning migrant children and young people. Their reasons for having migrated (e.g., economic or political) may vary, as well as their status—they may be citizens, residents, asylum seekers, refugees, unaccompanied minors or irregular migrants. Their length of stay in the host country may be short- or long-term, and they may or may not have the right to participate in the formal education system of the host country. Migrant children and young people from within and outside of the EU are taken into account" [27] (p. 11). For further details, visit the official webpage: https://eurydice.eacea.ec.europa.eu/publications/2019-eurydice-publications (accessed on 5 October 2023)

5   This framework to which we subscribe in this work incorporates a distinction between multilingualism, which is the social coexistence of different languages as separate systems in a given space or context; and plurilingualism, which is the individual's display of a unique repertoire of interrelated linguistic, cultural and multimodal resources [6] (p. 4). However, other authors (including some quoted in this paper) do not apply this distinction and might use bi- multi- or plurilingualism to describe similar phenomena and linguistic practices.

6   Informed consent was obtained from all subjects involved in the study. The interviews were conducted with the written authorization of the children's parents and the school management. The authorization documents had previously been validated by the NEW ABC project's team, which is dedicated to ethical procedures, and by the European Commission, as well as by the ethics committee of the Autonomous University of Barcelona.

7   For ethical reasons, all children's names were changed to pseudonyms.

8   All the fragments of data included in this analysis were translated by the authors into English.

9   See Note 7.

10   See Section 7 for a more detailed account of this particular case.

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
