# Peer review of "Exploring the Linguistic and Cultural Identities of Transnational Background Children in Catalonia, Spain"

_societies, doi:10.3390/soc13100221_

Round 1

Reviewer 1 Report

Despite the fact that this article is not offering something new in terms of promoting research in the fields of multilingualism and identities, it represents a well-conducted research and is very well presented. 

Author Response

Authors response to reviewer nº1:  

We appreciate the reviewer’s positive comments and evaluation. Regarding suggestions to improve the references, we have revised the article and final bibliography accordingly, correcting the format and also adding DOIs.

Reviewer 2 Report

The title is a bit clumsy and could be better formulated.

The article discusses an important topic of identity construction in an increasingly multicultural world resulting from transnational mobility.

The theory is well-developed, the author/s may like to mention also Glorii Anzaldúa’s nepantla/in-betweenness concept. On the other hand, however, while the theoretical introduction provides a useful background, it seems too long in proportion to the whole paper.

 The project is sufficiently extensively introduced.  

I would stress the value of the article in its goal of  raising “awareness among teachers, students and families about the benefits of plurilingual pedagogical approaches”, which  resonates with the needs of various other countries in Europe, faced with increasing influx of migrant and returning migrating families with children. Teachers in such an often new situation do need support, as the author/s rightly claim. 

The presented definition of a plurilingual speaker is highly useful in view of the increasing presence of plurilingualism on the European social scene.

The methodology as well as the course and results of the project are clearly described, while the findings very well organized and discussed.

I wonder whether popular children/youth culture the author/s refer/s to is predominantly American in origin? If yes, I think it should be noted.

Pupils’ identities are correctly identified and described. I would stress that considering the changing circumstances of migrants’ situation and the host communities, these identities are also not only complex, but also fluid The authors correctly support this notion and its significance for kids’ academic skills.

Author Response

Authors response to reviewer nº2:  

We appreciate the reviewer’s positive comments and agree with the valuable observations expressed in most of the points from the feedback. Regarding suggestions for improvement, we have revised the article accordingly. Here are some details of our revision: 

Reviewer’s comment: The title is a bit clumsy and could be better formulated. 

Changes:  

-We have replaced the original title (Reflecting on transnational background children’s linguistic and cultural identities in Catalonia, Spain) for a more concise and readable option (see page 1):  

Exploring the Linguistic and Cultural Identities of Transnational Background Children in Catalonia, Spain 

Reviewer’s comment: The theory is well-developed, the author/s may like to mention also Gloria Anzaldúa’s nepantla/in-betweenness concept. On the other hand, however, while the theoretical introduction provides a useful background, it seems too long in proportion to the whole paper. 

Changes:  

We have added the following text in relation to Anzaldúa’s Nepantla (see pages 9 and 10, lines 428-431):  

-One of the first authors to explore this phenomena was Anzaldúa (1987), who used the term ‘Nepantla’, a Nahuatl word, to refer to the space in between two worlds, where “you are not this or that” but actually “seeing from two or more perspectives simultaneously” (p.276). 

-References to Anzaldúa (1987) have also been included on pages 2 (line 69) and 9 (lines 385 and 423), and in the final bibliography, as: Anzaldúa, G. (1987). Borderlands/La Frontera. Aunt Lute Books. 

-We have also revised and edited the rest of the theoretical chapter for greater concision (see deleted parts in yellow across the text: pages 7, 8, 9 and 10).  

Reviewer’s comment: I wonder whether popular children/youth culture the author/s refer/s to is predominantly American in origin? If yes, I think it should be noted. 

Changes:  

-We agree with the reviewer in the relevance of noting the origin of these popular cultural expressions, which are in most cases either American or European. In this regard, we have included in their description a reference to their Western/mainstream origins and the need to critically reflect on this aspect too (see page 19, lines 796-797). 

All changes have been marked in yellow in the article for easier tracking.